# Comparison of the Clinical and Laboratory Findings and Outcomes of Hospitalized COVID-19 Patients Who Were Either Fully Vaccinated with Coronavac or Not: An Analytical, Cross Sectional Study

**DOI:** 10.3390/vaccines10050733

**Published:** 2022-05-07

**Authors:** Serap Şimşek Yavuz, Gülşah Tunçer, Özlem Altuntaş-Aydın, Mehtap Aydın, Filiz Pehlivanoğlu, Yeşim Tok, Sevim Mese, Alper Gündüz, Ceyda Geyiktepe Güçlü, İklima Özdoğan, Börçe Hemiş-Aydın, Pınar Soğuksu, Aysun Benli, Seniha Başaran, Kenan Midilli, Haluk Eraksoy

**Affiliations:** 1Department of Infectious Disease and Clinical Microbiology, İstanbul Faculty of Medicine, İstanbul University, İstanbul 34093, Turkey; r.borcehemis@hotmail.com (B.H.-A.); aysunsb@gmail.com (A.B.); seniha.basaran@istanbul.edu.tr (S.B.); heraksoy@gmail.com (H.E.); 2Department of Infectious Diseases and Clinical Microbiology, Haseki Training and Research Hospital, İstanbul 34093, Turkey; gulsah_durak_51@hotmail.com (G.T.); drfiliz@gmail.com (F.P.); ceydageyiktepe@gmail.com (C.G.G.); 3Department of Infectious Diseases and Clinical Microbiology, Çam and Sakura City Hospital, İstanbul 34093, Turkey; ozlemaa@gmail.com (Ö.A.-A.); alperg68@gmail.com (A.G.); 4Department of Infectious Diseases and Clinical Microbiology, Ümraniye Training and Research Hospital, İstanbul 34093, Turkey; mehtapaydin10@gmail.com (M.A.); iklima.ozdogan95@gmail.com (İ.Ö.); 5Department of Medical Microbiology, Cerrahpaşa Faculty of Medicine, Division of Virology, Istanbul Cerrahpaşa University, İstanbul 34093, Turkey; dr.yesimtok@gmail.com (Y.T.); kmidilli@gmail.com (K.M.); 6Department of Medical Microbiology, Istanbul Faculty of Medicine, Division of Virology and Fundamental Immunology, Istanbul University, İstanbul 34093, Turkey; drsevimmese@gmail.com (S.M.); pinar.soguksu@istanbul.edu.tr (P.S.)

**Keywords:** COVID-19, breakthrough COVID-19, inactivated SARS-CoV-2 vaccine, Coronavac

## Abstract

COVID-19 vaccines are highly protective against severe disease; however, vaccine breakthrough infections resulting in hospitalization may still occur in a small percentage of vaccinated individuals. We investigated whether the clinical and microbiological features and outcomes were different between hospitalized COVID-19 patients who were either fully vaccinated with Coronovac or not. All hospitalized COVID-19 patients who had at least one dose of Coronavac were included in the study. The oldest unvaccinated patients with comorbidities, who were hospitalized during the same period, were chosen as controls. All epidemiologic, clinical and laboratory data of the patients were recorded and compared between the fully vaccinated and unvaccinated individuals. There were 69 and 217 patients who had been either fully vaccinated with Coronavac or not, respectively. All breakthrough infections occurred in the first 3 months of vaccination. Fully vaccinated patients were older and had more comorbidities than unvaccinated patients. There were minor differences between the groups in symptoms, physical and laboratory findings, anti-spike IgG positivity rate and level, the severity of COVID-19, complications, and clinical improvement rate. The mortality rate of fully vaccinated patients was higher than the mortality rate in unvaccinated patients in univariate analysis, which was attributed to the fact that vaccinated patients were older and had more comorbidities. The severity and clinical outcomes of hospitalized patients with breakthrough COVID-19 after Coronavac vaccination were similar to those of unvaccinated patients. Our findings suggest that the immune response elicited by Coronovac could be insufficient to prevent COVID-19-related severe disease and death within 3 months of vaccination among elderly people with comorbidities.

## 1. Introduction

The Coronavirus Disease-2019 (COVID-19) pandemic began in Wuhan in December 2019 and has since affected more than 500,000,000 people, with 6 million dead all over the world so far. The major risk factors associated with death in patients with COVID-19 are older age (≥65 years), male sex, comorbidities including diabetes, chronic kidney, heart, lung and liver diseases, hypertension, obesity, cancer and immunocompromising states, the presence of severe disease with acute respiratory distress syndrome (ARDS), presence of neutrophilia, lymphopenia, higher serum D-dimer and ferritin levels on admission and being unvaccinated against the disease. The discovery of effective COVID-19 vaccines in as little as 11 months is one of the greatest achievements of medicine and scientific research in history. As of April 2022, the World Health Organization (WHO) has so far authorized a total of 14 severe acute respiratory syndrome coronavirus-2 (SARS-CoV-2) vaccines for COVID-19, in which 2, 3, 7 and 2 of them are mRNA, inactivated virus, viral vector (non-replicating) and protein subunit, respectively. After the administration of almost 8 billion doses of COVID-19 vaccines worldwide, it was understood that vaccines are highly effective in reducing COVID-19-related hospitalization, serious illnesses and death; however, the effectiveness of the vaccines vary by vaccine type and is sharply reduced over time; therefore, vaccine breakthrough infections resulting in hospitalization and death may occur in a small number of vaccinated, especially older and immunocompromised, patients [1,2,3,4]. It was recently reported that 2.6% of fully vaccinated elderly individuals had a breakthrough SARS-CoV-2 infection, 20% and 2.2% of breakthrough infections required hospitalization and resulted in death, respectively, and 34% of breakthrough hospitalizations required intensive care unit (ICU) admission [5]. Vaccinated people requiring hospital care due to severe COVID-19 breakthrough infection could have different clinical and laboratory features due to trained and previously acquired immunity. There are some reports of clinical and laboratory features of vaccine breakthrough infections after vaccinations with mRNA or adenoviral SARS-CoV-2 vaccines, but there is no article on the clinical characteristics of vaccine breakthrough infections after vaccination with Coronavac, which is an inactivated virus vaccine. In this study, we compared the clinical and microbiological features and outcomes of COVID-19 between Coronavac vaccinated and unvaccinated patients.

## 2. Material and Methods 

### 2.1. Study Design and Study Population

This was a multicenter, prospective, analytical cross-sectional study. 

Patients who had at least one dose of Coronovac and confirmed COVID-19 in the following days and were hospitalized in one of the five study hospitals from 01 April 2021 to 1 June 2021 were included in the study after obtaining informed consent. Patients who had two doses of Coronavac separated by at least 14 days and whose final dose was given at least 14 days earlier were accepted as fully vaccinated. For each vaccinated patient included in the study, one to two patients who were hospitalized in the same study hospital at the same time with the vaccinated cases due to confirmed COVID-19, had no COVID-19 vaccination previously and preferably with similar comorbidities and age were included as the control group. The oldest patients with comorbidities were chosen as the control. However, when the national vaccination campaign began it was aimed firstly at individuals older than 65 years, therefore it was impossible for us during the study period to find a control group with similar ages and comorbidities. All epidemiologic, clinical and laboratory data of the patients were recorded in previously prepared forms. The clinical severity of COVID-19 on admission was determined with the National Early Warning Score (NEWS) [6], and the progression of all patients was evaluated and compared with the “Ordinal Scale for Clinical Improvement” of the WHO [7], which was defined on admission and on days 3, 5, 7, 10, 14 and 21 of hospitalization. One nasopharyngeal swab and one plain tube of blood were taken, if possible, on the day of admission for the determination of SARS-CoV-2 RNA, variants of concerns (VOCs) and SARS-CoV-2 spike immunoglobuline G (IgG) antibody. Polymerase chain reaction (PCR) for SARS-CoV-2 RNA was performed on a Rotor-Gene Q 5 Plex Real Time (RT) PCR device (Qiagen, Germany) using a Bio-Speedy^®^ RT-qPCR detection kit (Bioeksen R&D Technologies Limited Company, İstanbul, Turkey). SARS-CoV-2 variant analysis was performed using a multiplex qRT-PCR assay (geneMAP^TM^ variant detection kit kindly provided by the manufacturer Genmark Sağlık Ürünleri, Istanbul, Turkey), which is designed for the detection and identification of VOCs. SARS-CoV-2 IgG antibodies were detected by enzyme linked immunosorbent assay ELISA using an anti-SARS-CoV-2 IgG kit (Euroimmun, Lübeck, Germany).

### 2.2. Statistical Analysis

The epidemiologic, clinical and laboratory features were compared between the fully vaccinated and unvaccinated patients. Comparisons were performed using the chi-square, Fisher’s exact, independent sample *t*-test or Mann–Whitney U tests, where appropriate. The variables were investigated using Kolmogorov–Smirnov test to determine whether or not they are normally distributed. A non-parametric Mann–Whitney U test was conducted to compare not-normally distributed parameters. Independent risk factors for mortality were defined by logistic regression analysis. IBM SPSS Statistics Version 21 was used for all of the statistical analysis. 

### 2.3. Ethics

The Ethical Committee of Istanbul University, Istanbul Faculty of Medicine approved the study on 9 July 2021, with a decision number of 14.

## 3. Results

### 3.1. Clinical and Laboratory Features of the Patients

A total of 292 patients who were hospitalized due to confirmed COVID-19 and were either Coronavac vaccinated or not vaccinated previously were included in the study. Six patients were removed as they were vaccinated with the Biontec-mRNA COVID-19 vaccine. While 175 (60.8%) of the patients had not received any SARS-CoV-2 vaccines, 111 had received at least one dose of Coronavac; 25 and 86 of the patients had received one and two shots of vaccination, respectively. Sixty-nine patients were accepted as fully vaccinated as they had received two shots, with the second dose being taken at least 14 days earlier. The mean age of the entire cohort was 66.44 ± 13.88 years, and 148 (51.4%) were female. All breakthrough infections occurred in the first 3 months of vaccination; the mean time from receipt of the second dose of the vaccine to the onset of COVID-19 was 38.42 ± 16.82 and the median was 38 (range 14–84) days; the 25th, 50th, 75th and 100th percentiles were 25, 38, 50 and 84 days, respectively.

### 3.2. Comparison of Vaccinated and Unvaccinated Patients

In total, 69 fully vaccinated patients were compared with 217 patients, either unvaccinated (175) or under-vaccinated (42). Fully vaccinated patients were older and more frequently had a sore throat, hypertension, chronic kidney disease, higher fever, higher blood pressure, higher serum creatinine and troponin levels and a higher hospital mortality rate than unvaccinated patients (*p* < 0.05, Table 1). We also compared 69 fully vaccinated patients with 175 unvaccinated ones and the results were very similar with the first analysis, except serum C reactive protein (CRP) level and white blood cell (WBC) number were significantly higher, and serum lactate dehydrogenase (LDH) level was significantly lower in fully vaccinated patients than unvaccinated ones; mean ± standard deviation (SD) values were 107 ± 69 mg/L versus 88 ± 66 mg/L, 9998 ± 11,021/µL versus 7835 ± 9998/µL and 346 ± 147 U/L versus 407 ± 232 U/L for CRP, WBC and LDH, respectively, among vaccinated versus unvaccinated patients. 

Unvaccinated patients more frequently had dyspnea, higher serum alanine aminotransferase (ALT) and ferritin levels, and higher blood lymphocyte counts on admission (*p* < 0.05, Table 1). NEWS scores were similar between the groups on admission and on days 3, 5, 7, 10, 14 and 21 of hospital stay. Additionally, laboratory findings on admission were similar between the vaccinated and unvaccinated patients, except vaccinated patients had higher blood WBC counts and serum troponin levels and unvaccinated patients had higher serum ALT and ferritin levels (*p* < 0.05). The results of the on-admission nasopharyngeal swab SARS-CoV-2 RNA PCR test and anti-SARS-CoV-2 Spike IgG were available for 94 and 74 patients, respectively. The nasopharyngeal swab SARS-CoV-2 RNA PCR test positivity rate and mean anti-SARS-CoV-2 spike antibody S/CO ratio on hospital admission were higher in vaccinated patients than in unvaccinated patients (*p* < 0.05). The cycle threshold (Ct) of the SARS-CoV-2 PCR test on hospital admission was similar between the groups. However, the Ct values were significantly higher among 46 anti-SARS-CoV-2-S antibody-positive patients than among 9 anti-SARS-CoV-2 -S antibody-negative patients (32.47 ± 4.180 vs. 25.314 ± 6.885) (*p* = 0.005). SARS-CoV-2 variant analysis was performed for 33 patients, 16 of whom were fully vaccinated and 17 of whom were unvaccinated. The alpha variant was the most frequently isolated variant whose presence rate was the same in the groups. Although the alpha+E484K variant was found to be more frequent (3/16 (18.8% vs. 1/17 (5.9%)) among patients with vaccine breakthrough infections, the difference was not statistically significant, probably due to the small number of patients. The presence of the E484K mutation was more frequent among fully vaccinated patients (4/16 (25%) vs. 2/17 (11.8)); however, the result also showed no statistical significance.

### 3.3. Analysis of Risk Factors for Mortality from COVID-19

In univariate analysis including all the patients, older age (*p* = 0.007), number of comorbidities (*p* = 0.007), chronic renal failure (*p* = 0.003), chronic obstructive lung disease (*p* = 0.009), heart disease other than coronary artery disease (congestive heart failure, heart valve diseases and hearth rhythm disorders) (*p* = 0.017), cancer (*p* = 0.017), full vaccination (*p* = 0.048), body temperature on admission (*p* = 0.011), haemoglobin (*p* = 0.021), procalcitonin (*p* = 0.001), creatinine (*p* = 0.009) and troponin (*p* = 0.001) levels, need for ICU (*p* < 0.001), oxygen support with high flow nasal cannula (HFNC) or non-invasive ventilation or mechanical ventilation (*p* < 0.001, for each), development of acute renal failure due to COVID-19 (*p* < 0.001), (*p* = 0.009) were found to be risk factors for mortality.

In a multivariate logistic regression analysis including all patients, the presence of chronic lung disease, cancer, chronic renal failure and other heart diseases, the need for oxygen with HFNC, and serum haemoglobin level on admission were found to be independent risk factors for mortality (Table 2).

The mean age of fully vaccinated patients was 75 years (ranging 53–92 years). Age, presence of chronic kidney disease, development of acute kidney injury, need for ICU care, HFNC, vasopressors and mechanical ventilation were risk factors for mortality among fully vaccinated patients. Although the Ct of the SARS-CoV-2 PCR test was higher for surviving patients (32.01 ± 5.18 in 29 surviving vs. 28.18 ± 8.01 in 4 deceased patients), the difference was not statistically significant.

The NEWS scores on admission and on day 3 of hospital stay were not different between the deceased and surviving fully vaccinated patients, but the NEWS scores on days 5, 7, 10, 14 and 21 of hospital stay were higher in deceased patients (*p* < 0.05) (Table 3)

## 4. Discussion

In this study, we found that patients hospitalized due to breakthrough COVID-19 after Coronavac vaccination were older and had more comorbidities than unvaccinated hospitalized COVID-19 patients. The reported features of vaccine breakthrough COVID-19 cases after mRNA and/or adenovirus vector-based vaccines were also quite similar to the features found in our study; among cases hospitalized for COVID-19, fully vaccinated ones were older and more likely to have underlying medical conditions or to be immunosuppressed compared with unvaccinated cases [3,8,9,10,11].

Our findings, along with others, show that older patients with comorbidities are among the most vulnerable population to severe vaccine breakthrough COVID-19. For this group of patients, close follow-up of vaccine response, administration of booster doses of vaccines if needed and the continuation of non-pharmaceutical interventions, including social distancing and mask wearing after vaccination, are quite important.

There were minor differences in the symptoms and physical findings of COVID-19 patients who were either fully vaccinated or unvaccinated in our study. Additionally, the laboratory findings on admission of vaccinated patients were similar to the findings of unvaccinated COVID-19 patients in our cohort, except that the mean ferritin level of vaccinated patients was lower than the levels in unvaccinated patients (690.04 ng/mL vs. 485.42 ng/mL). In another study from India, ferritin levels were also found to be lower in patients with vaccine breakthrough infections than in patients with primary infection (544.82 ng/mL vs. 392.26 ng/mL) [12]; however, no significant differences were found in white blood count, absolute lymphocyte count nadir, CRP, interleukin-6, procalcitonin, oxygen saturation, lung involvement, and fever frequency between the recipients of the first and second vaccine doses in a study conducted in Poland [13]. It should be known that the clinical course or laboratory findings of patients hospitalized due to breakthrough COVID-19 after Coronavac vaccination may not reveal any distinctive features. Despite the lower ferritin level, the severity of the disease, incidence of complications including acute kidney injury, requirements for intensive care unit support and mortality rate were not different between the vaccinated and unvaccinated patients in our study; NEWS scores of vaccinated and unvaccinated patients on admission were found to be 5.53 and 5.18, respectively. The mortality rate of fully vaccinated patients was found to be higher than the mortality rate of unvaccinated patients in the univariate analysis in our study, which was attributed to the fact that vaccinated patients were older and had more comorbidities. Indeed, comorbidities were found as independent risk factors for mortality in the multivariate analysis, while vaccination was not. In contrast to our study, the severity of the disease and a requirement for ventilator support were significantly lower in the vaccinated group than in the unvaccinated patients in the Indian study mentioned above; however, the mortality rate was not found to be significantly different between the groups in that study and, similar to our study, no significant differences were seen in the incidence of acute kidney injury, requirement for renal replacement therapy or thrombotic complications between the two groups [12]. In another study of 218 mRNA-vaccinated breakthrough delta variant COVID-19 cases hospitalized in a Singapore Hospital, despite the significantly older age of the vaccine breakthrough group, the odds of severe COVID-19 requiring oxygen supplementation were significantly lower following vaccination, and vaccination was found to be associated with a faster decline in viral RNA load and a robust serological response [14]. In a USA study of 67,311 laboratory-confirmed COVID-19-associated hospitalized adult cases, there were no differences in the risk for ICU admission or in-hospital death between 30,967 unvaccinated and 1,255 fully vaccinated persons [3]. Finally, in a recent study from the USA, among patients hospitalized with COVID-19, mRNA vaccine breakthrough cases less commonly received ICU care and invasive mechanical ventilation than unvaccinated cases, and the likelihood of death was also lower in the vaccinated cases; but these findings did not differ by age group or immunocompromised status [11]. These differences in the clinical severity and outcomes of vaccinated and unvaccinated hospitalized COVID-19 patients between our study and other cohorts might be associated with two reasons: One is that compared with the Indian and Singapore cohorts [12,14], our and USA’s [3] vaccinated breakthrough cases were older (mean ages were 75 and 73 vs. 58 and 56 years in our and USA and Indian and Singapore studies, respectively) and had more comorbidities (prevalence of HT+ DM were 72% + 44% and 70% + 39% vs. 51% vs. 7% + 19% in our and USA [3] and Indian and Singapore cohort [12,14], respectively). Both older age and comorbidities are well-defined risk factors leading to severe disease and lower antibody response and protection from vaccines or higher vaccine unresponsiveness. Coronavac was found to be 42% effective among elderly people (>70 years old) after ≥14 days of the 2nd dose; it declined with increasing age in a Brazilian study [15]. In another study from Poland, fully vaccinated breakthrough cases were more frequently confirmed to be vaccine non-responders [13]. Confirming this hypothesis, in a recent study from the UK, among 10,024 vaccinated individuals with SARS-CoV-2 infection (9479 of them were matched to unvaccinated controls), COVID-19 vaccination was found to be associated with a lower risk of several, but not all, COVID-19 sequelae in those with breakthrough SARS-CoV-2 infection, but these benefits of vaccination were clear in younger people but not in people over 60 years of age [16].

Another reason could be the type of vaccine used for vaccination. We included patients only vaccinated with Coronavac. Although all WHO-approved SARS-CoV-2 vaccines were shown to be effective against COVID-19, none of the COVID-19 vaccines created an equal level of protection, and the reported effectiveness of those vaccines against symptomatic COVID-19 varied from 50% to 95% between the vaccine types. mRNA vaccines were reported to be the most effective vaccines, followed by adenoviral vaccines and then inactivated vaccines [17]. Additionally, all vaccine types, including inactivated and mRNA vaccines, were shown to have lower efficacy among elderly people in real life [15,18].

It is widely accepted that protection from SARS-CoV-2 after vaccination is largely mediated by a neutralizing antibody (Nab) response, and there is a significant and positive correlation between serum anti-spike IgG levels and Nab titers [19,20]. Vaccine-elicited neutralization levels measured early after vaccination were found to be correlated with the subsequent protective efficacy measured in phase 3 trials [21], and the duration of immunity mainly resulted from the starting Nab levels. Due to the major differences in vaccine-elicited Nab levels between different types of vaccines, it was foreseen that the duration of protection induced by vaccines would also differ between the different vaccine types. Coronavac-elicited Nab titers were found to be lower than convalescent patients in a phase 2 study, and it was predicted that booster doses will be needed within 3 months after the 2nd dose; this time span was supposed to be 9–12 months for mRNA vaccines [22]. Serum Nab titers were lower after the second dose of vaccines in both Coronavac- and Biontech mRNA-vaccinated older (>65 years old) people than in younger people [15,22]. Additionally, the risk of symptomatic COVID-19 decreased with increasing levels of anti-spike IgG and anti-RBD IgG and Nab titer [23]. In real life, it was also seen that vaccine effectiveness decreases significantly over time. In an Israel study including almost 700,000 people, all of whom were vaccinated with the BioNTech/Pfizer mRNA BNT162b2 vaccine in a two-dose regimen, a significant correlation was found between time-from-vaccine and afforded protection against SARS-CoV-2 infection [24]. BNT162b2 mRNA vaccine-induced protection against SARS-CoV-2 infection was shown to wane rapidly following its peak after the second dose, but protection against hospitalization and death persisted at a robust level for 6 months after the second dose [25]. The efficacy of the Coronavac vaccine against the severe delta variant infection among elderly people was shown to be 79% 14–30 days after the second dose, and the efficacy decreased significantly 6–8 months later to 24% [26]. However, in our study, all breakthrough infections occurred during the first 3 months of vaccination, and the median time from the second dose to COVID-19 was 38.42 ± 16.82 days. As a result, it was concluded that not the decreasing level of Nabs but unresponsiveness or lower antibody response to the Coronavac vaccination is the main reason for the same rate of severe consequences of COVID-19 among fully vaccinated and unvaccinated cases in our cohort. As lower than 10% of our fully vaccinated cases had immunosuppressive treatment or conditions including SOT, BMT or active cancer chemotherapy in our cohort, the main reason for vaccine ineffectiveness could be immune senescence due to older age and other comorbidities. Our finding of the same rate of SARS-CoV-2 spike antibody positivity among fully vaccinated and unvaccinated patients also supports this idea. In a study of breakthrough COVID-19 after mRNA vaccinations, the frequency of detectable anti-Spike antibodies in vaccinated individuals was found to be higher (100%) than that in unvaccinated individuals (16%) [14].

Although the mean S/CO of anti-S IgG on the 7th day of complaints among Coronavac-vaccinated inpatients was higher than that among unvaccinated inpatients in our study, the numbers were found to be very close to one another, at 6.11 vs. 5.14, respectively. However, it was shown that SARS-CoV-2 breakthrough infections are a strong booster of the humoral immune response among fully vaccinated patients, the anti-spike IgG antibody levels were increased significantly as low as 2 to 4 days after the onset of symptoms of breakthrough infections, and the levels were more than 10-fold higher in fully vaccinated infected individuals than in unvaccinated uninfected individuals [12,27,28]. These findings also support the lower response rate to the Coronavac vaccine in our cohort.

Corbett et al. recently described the immune correlates of protection by mRNA SARS-CoV-2 vaccine in a primate model. They found that S-specific IgG thresholds for protection from pneumonia and URT infections were >336 IU/mL and >645 IU/mL, respectively [29]. Those numbers translate into 7.5 S/CO and 14.5 S/CO levels, respectively, in the EuroImmune kit that was used in our study [30]. Our mean anti-S IgG levels of 5 S/CO and 6 S/CO among unvaccinated and vaccinated patients, respectively, were lower than the estimated level of protection for pneumonia despite the rapidly boosting effect of breakthrough infections among vaccinated people [28]. However, a protective threshold level of anti-SARS-CoV-2 spike IgG is not fully defined at the moment; as a result, it is impossible to draw a final conclusion with these results, and studies determining the protective level of antibodies against symptomatic COVID-19 and COVID-19 pneumonia are urgently needed. This type of level would be useful, especially among elderly and immunocompromised persons in whom unresponsiveness to the vaccine is expected.

We found a higher percentage of SARS-CoV-2 PCR positivity on admission among vaccinated patients, which could also be a reflection of the older age and comorbidities of those patients. We found no significant difference in Ct values between vaccinated and unvaccinated groups infected with SARS-CoV-2. There were numerous studies with similar results, especially during the delta surge [31,32,33,34]. Given the substantial proportion of asymptomatic vaccine breakthrough cases with high viral loads, interventions, including masking and distancing, should be considered for all vaccinated persons especially in the settings with elevated COVID-19 transmission [35].

This study was performed during the alpha peak of the COVID-19 pandemic in Turkey and, as a result, most of our included patients were infected with the alpha variant. The E484 mutation, either with alpha mutations or by itself, was more frequently determined in vaccinated people in our study; however, it showed no statistical significance. The E484K mutation is known to reduce susceptibility to antibody-mediated neutralization and is more prone to vaccine escape, which is also supported by our results [36].

Although Coronavac vaccination has been shown to reduce COVID-19-related hospitalization [37,38], our study is the first to evaluate the association between vaccination with Coronavac and progression to critical illness among hospitalized COVID-19 patients.

Our study has some limitations. As we included only hospitalized COVID-19 patients, we were unable to describe the effect of Coronavac on reducing hospitalization among COVID-19 patients. Additionally, while we included all the Coronavac vaccinated hospitalized COVID-19 cases, we were not able to include all the unvaccinated COVID-19 patients due to the high number of patients because of the COVID-19 alpha surge in Turkey during the study period. This could result in selection biases of unvaccinated patients and could affect the results of the comparisons, but the inclusion of nearly three-times more unvaccinated patients than vaccinated patients could increase the power of our analysis. Although we included the oldest patients with comorbidities who were hospitalized at the same period as the vaccinated patients as control group, it was not possible for us to create similar vaccinated and unvaccinated groups in terms of age and comorbidities, since the vaccination process started with elderly and people with comorbidities in our country. However, to overcome this problem we performed a multivariate logistic regression analysis to define the independent risk factors for mortality.

## 5. Conclusions

Elderly people with comorbidities, who have been vaccinated with Coronavac should continue to practice non-pharmaceutical interventions, including social distancing and mask-wearing after vaccination, due to the ongoing risk of severe COVID-19, especially during uncontrolled infection transmission in the community. Our findings, including the occurrence of breakthrough COVID-19 in the first 3 months of vaccination, the same clinical and laboratory features and outcomes of COVID-19, the same rate of anti-spike IgG positivity, and minor differences in antibody levels among fully Coronavac vaccinated and unvaccinated patients suggest that the immune response elicited by Coronavac could be insufficient to prevent COVID-19-related hospitalization and death within 3 months of vaccination among elderly individuals with comorbidities. It also suggests that the problem is not decreasing immunity by time but the insufficient immune response to the Coronavac among this group of patients from the beginning of vaccination. Additional studies about the effectiveness of three or more doses of primary immunization or booster doses with Coronavac, heterologous vaccinations with different vaccine types or using more immunogenic vaccines rather than Coronavac should be conducted to find a solution for more improved protection of elderlies with comorbidities from COVID-19.

## Figures and Tables

**Table 1 vaccines-10-00733-t001:** Comparison of the clinical and laboratory features and outcomes of hospitalized COVID-19 patients who were either fully vaccinated with Coronavac or not.

Characteristics or Results	Patients not Vaccinated with Two Doses of Coronavac at Least 14 Days Earlier Total Number: 217 (Unvaccinated (175), Under-Vaccinated (42)).	Patients Vaccinated with Two Doses of Coronavac at Least 14 Days Earlier (n) Total Number: 69	*p*
Age(y) mean ± standard deviation (SD)	65.57 ± 13.95	75.63 ± 9.06	<0.001
Gender, female, n (%)	111 (51.2)	37 (53.6)	0.720
Duration of symptoms, day, mean ± SD	7.88 ± 3.69	6.95 ± 3.40	0.064
Symptoms			
Fever, n (%)	76 (35.2)	17 (24.6)	0.104
Shivering, n (%)	49 (22.6)	9 (13)	0.086
Cough, n (%)	141 (65)	48 (69.6)	0.483
Productive cough, n (%)	44 (20.3)	11 (15.9)	0.426
Fatigue, n (%)	146 (67.3)	48 (69.6)	0.723
Dyspnoea, n (%)	173 (79.7)	47 (68.1)	0.046
Sore throat, n (%)	26 (12)	16 (23.2)	0.022
Rhinorrhoea, n (%)	18 (8.3)	3 (4.3)	0.274
Anosmia, n (%)	28 (12.9)	10 (14.5)	0.735
Dysgeusia, n (%)	29 (13.4)	14 (20.3)	0.161
Headache, n (%)	58 (26.7)	20 (29)	0.714
Myalgia, arthralgia, n (%)	105 (48.4)	31 (44.9)	0.616
Nausea, vomiting, n (%)	41 (19)	20 (29)	0.078
Diarrhea, n (%)	21 (9.7)	6 (8.7)	0.808
Comorbidities			
Hypertension, n (%)	94 (43.3)	50 (72.5)	<0.001
Diabetes Mellitus, n (%)	72 (33.2)	31 (44.9)	0.077
Chronic lung diseases, n (%)	35 (16.1)	16 (23.2)	0.182
Chronic kidney disease, n (%)	15 (6.9)	12 (17.4)	0.010
Coronary artery disease, n (%)	41 (18.9)	17 (24.6)	0.301
Other heart disease, n (%)	24 (11.1)	13 (18.8)	0.096
Human Immunodeficiency Virus infection, n (%)	0 (0)	0 (0)	-
Solid Organ Transplantation, n (%)	3 (1.4)	2 (2.9)	0.403
Hematopoietic Stem Cell Transplantation, n (%)	0 (0)	1 (1.5)	0.239
Cancer, n (%)	17 (7.8)	9 (13)	0.190
Immunosuppressive treatment, n (%)	15 (6.9)	5 (7.2)	0.925
National Early Warning Score (NEWS) score on admission, mean ± SD	5.93 ± 2.90	5.48 ± 2.77	0.255
Presence of fever, n (%)	76 (35.2)	17 (24.6)	0.104
Temperature on admission, °C, mean ± SD	36.53 ± 0.56	36.74 ± 0.69	0.013
The highest temperature during the course of the disease, °C, mean ± SD	37.04 ± 0.67	36.97 ± 0.61	0.485
Duration of fever, mean ± SD	2.51 ± 1.53	1.60 ± 0.69	0.074
Respiratory rate on admission, mean ± SD	23.00 ± 4.38	22.71 ± 3.95	0.642
The highest respiratory rate during the disease course, mean ± SD	27.31 ± 7.36	26.26 ± 5.69	0.368
Heart rate on admission, mean ± SD	86.54 ± 15.61	82.57 ± 16.22	0.080
The highest heart rate during the disease course, mean ± SD	103.04 ± 19.34	101.01 ± 19.38	0.300
Blood pressure on admission, mean ± SD	121.18 ± 19.49	101.01 ± 19.38	0.370
The lowest blood pressure during disease course, mean ± SD	101.25 ± 17.00	106.54 ± 17.41	0.009
O_2_ saturation on admission (SpO_2_), mean ± SD	86.68 ± 7.29	89.23 ± 5.68	0.908
The lowest SpO_2_ during the disease course, mean ± SD	82.91 ± 11.24	85.40 ± 6.88	0.528
Partial O_2_ pressure (Pa O_2_)on admission, mean ± SD (n)	65.80 ± 28.11 (58)	68.11 ± 34.14 (18)	0.746
The lowest PaO_2_ during the disease course, mean ± SD (n)	55.58 ± 14.35 (60)	56.10 ± 17.86 (19)	0.899
Unilateral involvement on thorax computed tomography (CT), n (%)	17 (7.9)	5 (7.2)	0.851
Bilateral involvement on thorax CT, n (%)	194 (90.7)	61 (88.4)	0.586
50% involvement on thorax CT, n (%)	138 (64.5)	37 (53.6)	0.106
White blood cell (WBC) count on admission, mean ± SD	7852.58 ± 3 657.02	9998.26 ± 11 021.0	0.189
The highest WBC count during the disease course, mean ± SD	13,109.44 ± 6179.54	13,868.8 ± 6578.23	0.260
The lowest WBC count during the disease course, per microliter, mean ± SD	6261.15 ± 2706.30	6455.94 ± 3171.59	0.618
Blood haemoglobin level on admission, gr/dL, mean ± SD	12.45 ± 1.95	12.27 ± 1.90	0.500
The lowest blood haemoglobin level during the disease course, gr/dL, mean ± SD	10.99 ± 2.17	1.078 ± 1.92	0.455
Blood platelet count on admission, per microliter, mean ± SD	179,874 ± 123 599	163,791 ± 112 003	0.337
The lowest blood platelet count during the disease course, per microliter, mean ± SD	160,023 ± 115 627	138,031 ± 114 389	0.169
Blood polymorphonuclear leukocyte (PNL) count on admission, mean ± SD	6185.49 ± 3408.73	6826.38 ± 4365.70	0.521
The highest blood PNL count during the disease course, mean ± SD	10,897.58 ± 5791.07	12,463.8 ± 8927.46	0.246
Blood lymphocyte count on admission, mean ± SD	1133.845 ± 821.75	1088.99 ± 996.86	0.680
The lowest blood lymphocyte count during disease course, mean ± SD	729.46 ± 855.34	686.47 ± 1 194.41	0.015
Serum C-reactive protein (CRP) level on admission, mg/L, mean ± SD	92.77 ± 67.49	107.71 ± 69.85	0.076
The highest serum CRP level during the disease course, mg/L, mean ± SD	125.28 ± 87.96	143.23 ± 108.37	0.165
Serum procalcitonin level on admission, ng/µL, mean ± SD	0.43 ± 2.36 (190)	0.29 ± 0.46 (65)	0.295
The highest serum procalcitonin level during the disease course, ng/µL, mean ± SD	1.06 ± 5.23 (186)	1.27 ± 5.13 (64)	0.772
Serum alanine aminotransferase (ALT) level on admission, U/L, mean ± SD	40.19 ± 92.50	27.02 ± 24.32	0.099
The highest serum ALT level during the disease course, U/L, mean ± SD	197.98 ± 1024.87	62.39 ± 62.89	0.019
Serum aspartate aminotransferase (AST) level on admission, U/L, mean ± SD	46.20 ± 92.22	39.52 ± 26.20	0.896
The highest serum AST level during the disease course, U/L, mean ± SD	302.99 ± 2 160	66.76 ± 76.42	0.827
Serum creatinine level on admission, mg/dL, mean ± SD	1.04 ± 0.86	1.17 ± 0.61	0.021
The highest serum creatinine level during the disease course, mg/dL, mean ± SD	1.25 ± 1.12	1.33 ± 0.72	0.027
Serum glucose level on admission, mg/dL, mean ± SD	173.84 ± 97.14	151.27 ± 52.46	0.642
The highest serum glucose level during the disease course, mg/dL, mean ± SD	271.27 ± 128.43	259.10 ± 100.33	0.691
Serum ferritin level on admission, ng/mL, mean ± SD	690.04 ± 762.00	485.42 ± 512.46	0.010
The highest serum ferritin level during the disease course, ng/mL, mean ± SD	1206.61 ± 2360.71	1524.96 ± 4265.98	0.069
Serum D-dimer level on admission, mg/L, mean ± SD	2.07 ± 6.47	2.34 ± 4.98	0.321
The highest serum D-dimer level during the disease course, mg/L, mean ± SD	3.92 ± 6.49	3.97 ± 6.49	0.358
Serum troponin level on admission, pg/mL, mean ± SD (n)	21.18 ± 45.92 (198)	28.61 ± 42.60 (59)	<0.001
The highest serum troponin level during the disease course, pg/mL, mean ± SD (n)	37.72 ± 107.03 (201)	76.84 ± 254.90 (61)	0.083
Serum lactate dehydrogenase (LDH) level on admission, U/L, mean ± SD	396.55 ± 219.31	404.42 ± 515.57	0.077
The highest serum LDH level during the disease course, U/L, mean ± SD	722.34 ± 2620.94	467.36 ± 241.56	0.061
Serum interleukin-6 (IL-6) level on admission, mean ± SD (n)	72.72 ± 164.26 (20)	32.77 ± 50.08 (6)	0.242
The highest serum IL-6 level during the disease course, mean ± SD (n)	94.15 ± 96.51 (11)	125.95 ± 11.24 (2)	0.662
Myocarditis, n (%)	0	0	-
Acute kidney injury, n (%)	12 (5.5)	4 (5.8)	1.000
DIC, n (%)	0	0	-
O_2_ supplement with nasal cannula, n (%)	171 (78.8)	49 (71)	0.181
O_2_ supplement with face mask with reservoir, n (%)	133 (61.3)	43 (62.3)	0.878
O_2_ supplement with high flow nasal cannula, n (%)	46 (21.2)	13 (18.8)	0.673
Non-invasive ventilation, n (%)	12 (5.5)	8 (11.6)	0.085
Mechanic ventilation, n (%)	16 (7.4)	9 (13)	0.146
Need for intensice care unit support, n (%)	37 (17.1)	13 (18.8)	0.733
Need for vasopressors, n (%)	12 (5.5)	6 (8.7)	0.346
Need for extracorporeal membrane oxygenation support, n (%)	0	0	-
Need for continuous renal replacement therapy, n (%)	4 (1.8)	2 (2.9)	0.594
Total duration of favipiravir treatment, day, mean ± SD	8.47 ± 2.57	8.56 ± 2.17	0.803
Glucocorticoid use for COVID-19, n (%)	198 (91.2)	63 (91.3)	0.988
Total administered dose of dexamethasone for COVID-19, mg, mean ± SD	37.79 ± 33.26	42.54 ± 37.76	0.448
Total administered dose of prednisolone for COVID-19, mg, mean ± SD	1012.9 ± 829.71	1202.80 ± 926.97	0.161
Anti-cytokine treatment for COVID-19, n (%)	36 (16.6)	8 (11.6)	0.316
Tocilizumab, n (%)	18 (8.3)	2 (2.9)	0.126
Anakinra, n (%)	21 (9.7)	6 (8.7)	0.808
Score of World Health Organization (WHO) ordinal scale on admission, mean ± SD (n)	3.42 ± 0.98 (214)	3.435 ± 0.9774 (69)	0.674
Score of WHO ordinal scale on day 3, mean ± SD (n)	3.93 ± 0.64 (216)	3.95 ± 0.49 (69)	0.674
Score of WHO ordinal scale on day 5, mean ± SD (n)	3.84 ± 0.89 (216)	3.94 ± 0.87 (68)	0.654
Score of WHO ordinal scale on day 7, mean ± SD (n)	3.64 ± 1.17 (214)	3.71 ± 1.29 (66)	0.654
Score of WHO ordinal scale on day 10, mean ± SD (n)	3.17 ± 1.52 (209)	3.27 ± 1.61 (65)	0.734
Score of WHO ordinal scale on day 14, mean ± SD (n)	2.67 ± 1.64 (203)	2.93 ± 1.83 (62)	0.393
Score of WHO ordinal scale on day 21, mean ± SD (n)	2.15 ± 1.78(192)	2.32 ± 1.72 (56)	0.307
Nasopharyngeal swab SARS-CoV-2 PCR positivity on hospital admission, n (%)	42/58 (72.4)	33/36 (91.7)	0.033
Cycle threshold of SARS-CoV-2 PCR test, mean ± SD	32.07 ± 5.33	31.55 ± 5.58	0.340
Distribution of variants	15/17 (88.2)	12/16 (75)	
Alpha, n (%)	1/17 (5.9)	3/16 (18.8)	0.521
Alpha+E484K, n (%)Others, n (%)	1/17 (5.9)	1/16 (6.3)	
Presence of E484K mutation, n (%)	2/17 (11.8)	4/16 (25%)	0.398
Anti Spike antibody, S/CO, mean ± SD (n)	5.14 ± 2.57 (41)	6.11 ± 3.19 (33)	0.025
Anti S antibody positivity, n (%)	36/41 (87.8%)	28/33 (84.8%)	0.712
Mortality, n (%)	17 (7.8%)	11 (15.9%)	0.048

**Table 2 vaccines-10-00733-t002:** Results of multivariate analysis of mortality risk factors for COVID-19.

Variants	*p*	OR	95% Confidence Interval (CI)	
O_2_ supplement with high flow nasal cannula, n (%)	<0.001	6.254	2.230	17.534
Blood haemoglobin level on admission	0.041	0.772	0.603	0.989
Presence of chronic lung diseases	0.031	2.999	1.108	8.111
Presence of cancer	0.022	4.345	1.234	15.296
Presence of chronic renal failure	0.014	4.049	1.335	12.283
Presence of cardiac diseases other than coronary heart disease	0.003	6.017	1.819	19.904

**Table 3 vaccines-10-00733-t003:** Comparison of the patients who recovered or died after vaccine breakthrough COVID-19.

Characteristics or Results	Patients Fully Vaccinated with Coronavac n: 69	Fully Vaccinated Patients Who Were Recovered After COVID-19 n: 58	Fully Vaccinated Patients Who Have Died after COVID-19 n: 11	*p*
Age, mean ± standard deviation (SD)	75.63 ± 9.06	74.53 ± 9.18	81.45 ± 5.76	0.019
Gender, female, n (%)	37 (53.6)	34 (58.6)	3 (27.3)	0.097
Days after the second dose of the vaccine, mean ± SD	38.42 ± 16.82	33.18 ± 14.32	39.41 ± 17.18	0.263
Comorbidities				
Hypertension, n (%)	50 (72.4)	41 (70.7)	9 (81.8)	0.715
Diabetes Mellitus, n (%)	31 (44.9)	29 (50)	2 (18.2)	0.095
Chronic lung diseases, n (%)	16 (23.2)	11 (19)	5 (45.5)	0.056
Chronic kidney disease, n (%)	12 (17.4)	7 (12.1)	5 (45.5)	0.007
Coronary artery disease, n (%)	17 (24.6)	16 (27.6)	1 (9.1)	0.270
Human Immunodeficiency Virus infection, n (%)	0 (0)	0 (0)	0 (0)	-
Solid Organ Transplantation, n (%)	2 (2.9)	2 (3.4)	0 (0)	1.000
Hematopetic Stem Cell Transplantaion, n (%)	1 (1.45)	1 (1.7)	0 (0)	1.000
Cancer, n (%)	9 (13)	8 (13.8)	1 (9.1)	1.000
Immunosuppressive treatment, n (%)	5 (7.3)	5 (8.6)	0 (0)	0.585
NEWS score on admission, mean ± SD	5.48 ± 2.77	5.53 ± 2.69	5.18 ± 3.28	0.702
O_2_ saturation on admission, mean ± SD	89.23 ± 5.68	89.12 ± 5.78	89.88 ± 5.30	0.732
Bilateral involvement on thorax computed tomography CT, n (%)	61 (88.40)	52 (89.7)	9 (81.8)	0.604
>50% involvement on thorax CT, n (%)	37 (53.6)	31 (53.4)	6 (53.5)	0.947
Anti Spike Sample/Cut-off ratio, mean ± SD	6.11 ± 3.19 (33)	6.53 ± 2.99(28)	3.76 ± 3.48 (5)	0.034
Myocarditis, n (%)	0 (0)	0	0	-
Acute kidney injury, n (%)	4 (5.8)	1 (1.7)	3 (27.3)	0.011
Disseminated intravascular coagulation, n (%)	0 (0)	0	0	-
O_2_ supplement with nasal cannula, n (%)	49 (71.0)	42 (72.4)	7 (63.6)	0.556
O_2_ supplement with reservoir face mask, n (%)	43 (62.3)	35(60.3)	8(72.7)	0.437
O_2_ supplement with high flow nasal cannula, n (%)	13 (18.8)	6 (10.3)	7 (63.6)	<0.001
Mechanic ventilation, n (%)	9 (13.0)	2 (3.4)	7 (63.6)	<0.001
Need for intensive care unit, n (%)	13 (18.8)	4 (6.9)	9 (81.8)	<0.001
Vasopressors, n (%)	6 (8.7)	1 (1.7)	5 (45.5)	<0.001
Total duration of favipiravir treatment, day, mean ± SD	8.56 ± 2.17	8.36 ± 2.26	9.63 ± 1.20	0.074
Glucocorticoid use for COVID-19, n (%)	63 (91.3)	52 (89.7)	11 (100)	0.580
Total administered doses of dexamethasone for COVID-19, mg, mean ± SD	42.54 ± 37.76	44.82 ± 41.25	34.80 ± 24.14	0.704
Total administered doses of prednisolone for COVID-19, mg, mean ± SD	1202.80 ± 926.97	1138.64 ± 921.67	1502.22 ± 945.75	0.290
Anti-cytokine treatment for COVID-19, n (%)	8 (11.6)	5 (8.6)	3 (27.3)	0.109
Tocilizumab, n (%)	2 (2.9)	2 (3.4)	0 (0)	1.000
Anakinra, n (%)	6 (8.7)	3 (5.2)	3 (27.3)	0.047
Score of World Health Organization (WHO) ordinal scale on admission, mean ± SD	3.46 ± 0.90	3.46 ± 0.88	3.45 ± 1.03	0.831
Score of WHO ordinal scale on day 3, mean ± SD	3.95 ± 0.49	3.98 ± 0.44	3.81 ± 0.75	0.598
Score of WHO ordinal scale on day 5, mean ± SD	3.94 ± 0.87	3.82 ± 0.79	4.60 ± 1.07	0.029
Score of WHO ordinal scale on day 7, mean ± SD	3.71 ± 1.29	3.44 ± 1.15	5.20 ± 1.03	<0.001
Score of WHO ordinal scale on day 10, mean ± SD	3.27 ± 1.61	2.90 ± 1.40	5.30 ± 1.15	<0.001
Score of WHO ordinal scale on day 14, mean ± SD	2.93 ± 1.83	2.38 ± 1.37	5.80 ± 1.13	<0.001
Score of WHO ordinal scale on day 21, mean ± SD	3.32 ± 1.72	1.83 ± 1.12	5.71 ± 1.38	<0.001

## Data Availability

All of the raw data of the included patients are available on google form through this link: https://docs.google.com/spreadsheets/d/15g8DFOJrVeBASSz8sIe32ILlsZihoepkxx0Dexh3x70/edit?usp=sharing. Accessed date: 1 April 2022. Forms that were used to collect the data for every patient and SPSS files for statistical analysis are also available upon request.

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
