# Peer review of "Comparison of the Clinical and Laboratory Findings and Outcomes of Hospitalized COVID-19 Patients Who Were Either Fully Vaccinated with Coronavac or Not: An Analytical, Cross Sectional Study"

_vaccines, 2022, doi:10.3390/vaccines10050733_

Round 1

Reviewer 1 Report

Overall it is a clear and important work. I would rather advise the authors to improve their English grammar and style.
All in all, I recommended acceptance after minor revision.

Author Response

Dera Reviewer,

Thank you very much for your contributions. I attached my responses to your comments. Kind regards. 

Reviewer 2 Report

Dear Authors, 
It was a pleasure to be appointed as the reviewer for the article ˝Comparison of the Clinical and Laboratory Findings and Out-2 comes of Hospitalized COVID-19 Patients Who were either 3 Fully Vaccinated with Coronavac or not: An Analytical, Cross 4 Sectional Study˝. Data regarding inactivated vaccines are lacking, and it was interesting to see the results of your study.

My major concern is the study design.

The study description states that control subjects were selected among the unvaccinated COVID-19 patients to be similar in age and comorbidities to vaccinated COVID-19 subjects. It is a surprise to have a statistically significant difference in age and comorbidities between the control and vaccinated groups, as is stated in the results.

Additionally, fully vaccinated subjects were compared with a mixed control group of once, twice and not vaccinated subjects. A comparison of not vaccinated at all and fully vaccinated groups is missing. 
Is there a possibility to compare fully vaccinated with the all patients admitted to the hospitals? Or at least randomly selected?

It is troublesome to have a specially selected control group significantly different in age, chronic renal disease and blood pressure, all known to be the risk factors for COVID-19 outcome. 
A new control group will make it possible to assess age and comorbidities as the risk factors. 

Author Response

Dear Reviewer,

Thank you very much for your precious contributions and recommendations. I attached the file including our responses to your recommendations. Kind regards. 

Reviewer 3 Report

This is a very interesting paper investigating clinical and laboratory findings and outcomes of hospitalized patients suffering from COVID-19 infection in patients having received Coronavac or not. The paper is well-written, and of interest for the readers. However, several changes should be made before publishing it.

In the abstract section  the authors are described the methods. They compared hospitalized patients with covid-19 infection who received the Coronavac vaccine and "some" unvaccinated patients. What does "some" mean? Which was the method of selection of such control group? It should be clarified in the methods section.

In the first part of the introduction, the authors are introducing the vaccination against covid. I would recommend to start with a brief introduction of the COVID pandemic, and a brief process prior to the development of vaccines.

The introduction is really short. It should be expanded. How many vaccines ara available? They should be briefly described.

The material and methods section can be divided into several subsections: study population, study design, ethics and statistical analysis.

The results section can be also divided into several subsections according to the results reported. In a first part, the authors can describe analyses concerning the clinical and laboratory features of patients. This is corresponding to the Table 1. In a second part, mortality risk factors, and finally, the authors can separately describe differences between COVID-19 patients who recovered or died after COVID-19.

The conclusions section is brief. It should be expanded and open to the questions that remain still unresolved. The authors should compare their results with those from studies analyzing other vaccines. 

A future perspective subsection is needed. 

Author Response

(The authors gave the same response as above.)
